**Subject Category:**
Biology (whole organism)

biomechanics/physiology/environmental science

mysticete, keratin, filter-feeding, adsorption, pollution, conservation

**Author for correspondence:**
Alexander J. Werth
e-mail: awerth@hsc.edu

# Oil adsorption does not structurally or functionally alter whale baleen

Alexander J. Werth, Shemar M. Blakeney
and Adrian I. Cothren

Department of Biology, Hampden-Sydney College, Hampden-Sydney, VA 23943, USA

AJW, 0000-0002-7777-478X

Mysticete whales filter small prey from seawater using baleen, a unique keratinous oral tissue that grows from the palate, from which it hangs in hundreds of serial plates. Laboratory experiments testing effects of oils on material strength and flexibility, particle capture and tissue architecture of baleen from four mysticete species (bowhead, *Balaena mysticetus*; North Atlantic right, *Eubalaena glacialis*; fin, *Balaenoptera physalus*; humpback, *Megaptera novaeangliae*) indicate that baleen is hydrophilic and oleophobic, shedding rather than adsorbing oil. Oils of different weights and viscosities were tested, including six petroleum-based oils and two fish or plankton oils of common whale prey. No notable differences were found by oil type or whale species. Baleen did not adsorb oil; oil was readily rinsed from baleen by flowing water, especially from moving fringes. Microscopic examination shows minimal wrinkling or peeling of baleen's cortical keratin layers, probably due to oil repelling infiltrated water. Combined results cast doubt on fears of baleen fouling by oil; filter porosity is not appreciably affected, but oil ingestion risks remain. Particle capture studies suggest potentially greater danger to mysticetes from plastic pollution than oil.

## 1. Introduction

Oil poses serious health risks to baleen whales. It irritates skin, eyes and mucous membranes, increasing susceptibility to infection, and when ingested inflames or damages the gastrointestinal tract, respiratory system and internal organs [1–3]. Fouling of baleen—the unique keratinous oral tissue on which mysticetes depend for filter feeding—is another major concern; it could increase risk of oil ingestion and decrease filtering efficiency [3–5], the subject of intense speculation but limited experimentation.

However, the generic term 'oil' encompasses diverse non-polar substances, including not just petroleum products but also biogenic, animal-derived lipids common in fish and

**Figure 1.** (a) Right whales swim through and may seek oil slicks from planktonic prey patches (credit: Center for Coastal Studies, #3691 on 4-25-2007-N, NMFS Permit #633-1763-00). (b) bowhead whale stomach contents, of which 20−25% are typically pure copepod oil, show a swirled, reflective sheen when poured into water (credit: T. Sformo & J.C. George).

zooplankton. Copepods can store half their mass in oil chemically similar to fish oil [6]. In fact, fish do not synthesize omega-3 fatty acids but obtain them from dietary zooplankton [7]. Although this study was initially undertaken to investigate environmental interactions between baleen and petroleum-based hydrocarbon oils, it expanded to include organic oils abundant in mysticete diets [8].

We focused on bowhead (*Balaena mysticetus*) and right (*Eubalaena* spp.) whales, particularly the North Atlantic right whale (*Eubalaena glacialis*), for four reasons. First, whales of the family Balaenidae have exceptionally long baleen plates (greater than 4 m) [9] and hence the greatest overall filtering surface area among all mysticetes [10]. Second, balaenids' slow, continuous skim feeding, often at the surface of the water column [11,12], puts them at great risk of oil ingestion. Third, balaenids are already highly endangered species [13]. Fourth, they are likely to encounter oil in native habitats. This relates to geographical distribution—for the circumpolar bowhead, near natural petroleum seeps, offshore platforms and tankers; for migrating or feeding right whales, near commercial ports and shipping lanes—but also because balaenid whales typically feed on oil-rich copepods [14]. Dense patches of copepods can be located by human observers, and perhaps by whales, via oily surface slicks (figure 1a). Bowhead stomach contents [15] reveal large quantities of liquid oil (figure 1b). For comparison, we also studied baleen of humpback (*Megaptera novaeangliae*) and fin (*Balaenoptera physalus*) whales, with a lipid-rich diet from krill and forage fish [8,12].

Baleen is an anisotropic tissue, with a pair of flat cortical sheets enclosing hair-like tubules and intertubular, medullary keratin [10,16,17]. Baleen is ever-growing at its roots in the palatal gingiva [18], but where exposed comprises dead, cornified cells and keratin fibres [16,19]. Baleen is flexible and strong when hydrated, as *in vivo* [17,20]. However, the potential effects of oil on baleen structure and function are largely unknown despite the risks of baleen encountering petroleum-based oil and the high likelihood of baleen encountering biogenic, prey-based oil.

In this integrative, multifactorial study, we synthesized four independent lines of controlled, laboratory-based investigation to determine potential differences in oiled versus unoiled baleen. In terms of *physical properties*, we tested baleen's adsorptive ability for oil (i.e. how well oil adhered to the baleen surface). We examined baleen's *histological properties* by searching for microscopic tissue changes due to oil. We investigated *mechanical properties* via material testing of composite flexure of baleen. Finally, we explored *functional properties* by studying baleen's capture of particles via flow tank experiments.

Our null hypothesis was that exposure to (and in particular, long-term immersion in) oils will not affect baleen structure or function. Our alternative hypothesis was that oil will in fact affect baleen's physical, histological, mechanical and functional properties outlined above.

# 2. Material and methods

## 2.1. Specimens

Baleen from four species was studied: bowhead (*Balaena mysticetus*) and North Atlantic right whales (*Eubalaena glacialis*) of Balaenidae, and fin (*Balaenoptera physalus*) and humpback whales (*Megaptera*

*novaeangliae*) of Balaenopteridae, commonly called rorquals. No animals were killed or harmed to collect baleen; no baleen was imported from outside USA. Bowhead baleen was obtained from Inupiat subsistence hunters in Utqiaġvik (Barrow), Alaska; all other specimens came from animals that died naturally prior to or during stranding in five states along the US Atlantic coast, with collection by the NOAA/NMFS Northeast Marine Mammal Stranding and Disentanglement Network (Virginia, Massachusetts) or NOAA/NMFS Southeast Marine Mammal Stranding Network (North Carolina, Florida, Georgia). All baleen specimens were frozen for shipment, then thawed at room temperature or in chilled water for storage and testing.

## 2.2. Experimental design and statistical power analysis

Given the complexity of our experimental set-up—testing baleen from four species in multiple ways (oil adsorption, mechanical strength, particle capture, etc.) following exposure to eight types of oil—we conducted a basic power analysis prior to data collection to ensure that our null hypothesis testing would be valid and robust. Using WebPower 4.5.2 (an online R package [21]), we determined the minimum sample size needed to avoid Type II (false negative) errors due to insufficient statistical power: for the oil adsorption testing, $n = 10$ (for each oil versus non-oil; $n = 340$ total) with 0.80 power ($\pi$), representing greater than 80% confidence for significance at the 0.05 level. This was the same ($n = 340$) for the mechanical testing. For the histology analysis, the minimum $n = 35$ samples per species. For the functional (flow tank particle capture) testing, we had a minimum $n = 12$ trials per flow speed (total $n = 180$).

Data from each component of the study were analysed statistically via paired *t*-tests to compare different (air versus water) pre-treatments from the same species and ANOVA to compare results of exposure to the different oils.

## 2.3. Oil adsorption experiments

To study oil absorption, baleen from each species was cut into $5 \times 5$ cm including medial fringes. Half of the squares were kept for 7 days in air at $22°C$ and half for 7 days in flowing $17°C$ seawater, then all squares were submerged an additional 7 days in undiluted oil. We used eight oils (table 1) ranging from lightweight, low-viscosity machine, lubricant and mineral oils to heavier SAE-30 chain oil, SAE-40 motor oil and GL-4 gear oil (density $0.81–0.93$ g cm$^{-3}$), plus natural cod liver and albacore tuna oils ($0.92$ g cm$^{-3}$). Crude oil was not used; it is highly volatile, with flammable vapours and toxic outgassing. We used refined hydrocarbon alkanes separated from crude oil via fractional distillation. Testing of heavier fuel oils (and chemical dispersants) is ongoing.

Five baleen samples were used of each mysticete species per treatment. There were 16 different oil treatments including eight oils with pre-oil storage for a week in air or water, plus control samples kept for two weeks in seawater alone (=17 total scenarios). Thus, a total of 85 samples were used for each species, or 340 baleen samples overall. We used additional $5 \times 5$ cm of aluminium and HDPE plastic as controls for each oil and pre-oil air or water treatment. Oils were kept in glass beakers or bowls at room temperature ($21–22°C$). Baleen and control material samples were laid flat on the bottom of each container and turned over once, after 4 days, so that each surface of a flat square would be more readily exposed to oil. Oil was not mixed or otherwise disturbed during the week in which the squares were submerged.

To judge how well oils were adsorbed by (i.e. adhered to) the sample squares, we weighed them with an electronic balance. Before weighing, samples were held upright for 20 s following removal from the oil container to allow accumulated surface oil to drip off, but in no cases were samples wiped, rinsed or otherwise cleaned. After recording weights, we analysed data statistically, using paired *t*-tests to compare data from the same species and oil treatment with different (air versus water) pre-treatment. We then used ANOVA to compare results of exposure to the eight different oils.

## 2.4. Histological examination

Following removal from oil containers (for adsorption testing), $5 \times 5$ cm baleen squares of each species and oil treatment were examined (and compared to control samples of baleen stored in air or water) to determine (i) the extent to which oil(s) were adsorbed at fine scale and (ii) if tissues were transformed or damaged by full immersion in oil for 7 days. This investigation focused on the integrity of the flat, sheet-like cortical keratin layers on the flat anterior and posterior faces of each baleen plate. Samples were

**Table 1.** Mean ± s.d. relative change in mass [1.0 = no change] of baleen samples with 7 day exposure to oil. For comparison, the first (left) column shows change in mass of baleen samples with 7 day submersion in water alone. Other columns show mass change for dried or wet baleen samples (i.e. kept in air or water for 7 days) prior to being submerged 7 days in various oils (Ch = SAE-30 chain, Ge = GL-4 gear, Mo = SAE-40 engine motor, Lu = multipurpose lubricating, Mi = mineral, Al = alkanes, Co = cod liver, Tu = tuna). Baleen gains more mass from water than from any oil.

| species | water | dried baleen (in air 7 days before oil exposure) | | | | | | | | wet baleen (in water 7 days before oil exposure) | | | | | | | |
|---|---|---|---|---|---|---|---|---|---|---|---|---|---|---|---|---|---|
| | $H_2O$ | Ch | Ge | Mo | Lu | Mi | Al | Co | Tu | Ch | Ge | Mo | Lu | Mi | Al | Co | Tu |
| **N. Atl. right whale** | 1.29 | 1.08 | 1.07 | 1.08 | 1.09 | 1.10 | 1.08 | 1.13 | 1.14 | 1.14 | 1.14 | 1.16 | 1.15 | 1.21 | 1.12 | 1.19 | 1.19 |
| *Eubalaena glacialis* | 0.08 | 0.05 | 0.06 | 0.05 | 0.04 | 0.08 | 0.03 | 0.06 | 0.04 | 0.07 | 0.04 | 0.06 | 0.06 | 0.08 | 0.03 | 0.04 | 0.05 |
| **bowhead whale** | 1.30 | 1.04 | 1.05 | 1.05 | 1.09 | 1.10 | 1.11 | 1.12 | 1.11 | 1.15 | 1.18 | 1.18 | 1.17 | 1.20 | 1.15 | 1.18 | 1.20 |
| *Balaena mysticetus* | 0.06 | 0.06 | 0.04 | 0.07 | 0.06 | 0.07 | 0.08 | 0.06 | 0.05 | 0.05 | 0.04 | 0.05 | 0.05 | 0.08 | 0.04 | 0.06 | 0.06 |
| **humpback whale** | 1.34 | 1.09 | 1.09 | 1.07 | 1.12 | 1.11 | 1.12 | 1.13 | 1.12 | 1.16 | 1.17 | 1.17 | 1.19 | 1.20 | 1.11 | 1.17 | 1.18 |
| *Megaptera novaeangliae* | 0.08 | 0.07 | 0.07 | 0.07 | 0.05 | 0.06 | 0.05 | 0.06 | 0.09 | 0.08 | 0.06 | 0.08 | 0.04 | 0.06 | 0.06 | 0.07 | 0.08 |
| **fin whale** | 1.36 | 1.10 | 1.09 | 1.08 | 1.11 | 1.11 | 1.10 | 1.14 | 1.14 | 1.17 | 1.17 | 1.16 | 1.18 | 1.19 | 1.14 | 1.17 | 1.21 |
| *Balaenoptera physalus* | 0.09 | 0.07 | 0.08 | 0.08 | 0.05 | 0.07 | 0.06 | 0.07 | 0.08 | 0.09 | 0.09 | 0.08 | 0.06 | 0.07 | 0.07 | 0.09 | 0.08 |

viewed at 1–5× magnification through an Olympus SZH-10 stereoscopic dissecting microscope, or at higher magnifications (15–40×) with a Nikon Alphaplot YS2 microscope, and photographed with a 6MP OptixCam digital microscope camera and OCView software (Roanoke, Virginia, USA). In no case was there any standard histological treatment (e.g. embedding, sectioning or staining) of baleen: samples were examined directly (within 5 min) after removal from oil without any wiping, rinsing or blotting of the baleen surface.

During microscopic analysis, attention was paid to the amount of oil that was visible on the baleen surface. For at least one sample per treatment and species, a photograph was taken and analysed (using ImageJ, NIH, Bethesda, Maryland, USA) to determine the percentage of the surface area covered with visible oil residue. Additionally, notes and photographs were taken to record the potential presence of any surface irregularities (compared to baleen kept in air or seawater but not exposed to oil) such as cracking, pitting, peeling, wrinkling or splitting of keratin surface layers, and separation of the surface from deeper, underlying layers of keratin.

## 2.5. Mechanical (strength) testing

For mechanical testing, additional 5 × 5 cm of baleen from all four whale species were subjected to oil treatments as outlined above (using many of the same specimens used in the initial adsorption and weight experiments). Following removal from oil containers, the compressive strength of baleen squares was investigated via three-point bending tests using a Mark-10 ES30 universal testing machine with M4–200 force gauge (Copiague, New York, USA). Samples were patted dry with a paper towel to remove excess surface oil (if any), then placed on the testing machine's two metal arms spaced 24 mm apart. Samples were all placed in the same orientation, with baleen's internal horn tubes or emergent fringes placed perpendicular to the direction of these arms.

The Mark-10 testing device recorded maximal force (in N) encountered before sample failure (i.e. cracking) or when the machine reached a 30 mm displacement limit without failure. Using the thickness of the baleen ($\bar{x} = 2.84$ mm for all species, s.d. = 0.17, range 2.62–3.15 mm, $n = 336$) and the distance between the arms, flexural stress and strain were computed for each trial. From these measures, flexural stiffness (a function of the modulus) could be determined. Because stress–strain curves were not linear, flexural stiffness was defined as the largest slope regressed with a continuous subset of 25% original data [17,20]. After mechanical tests were performed for samples from all treatments with all species (including baleen kept for 7 days in air or flowing water before the 7 day period of immersion in various oils), strength testing was repeated on aluminium and HDPE plastic control squares plus control baleen that was not oiled. As before, results were analysed via paired *t*-tests and ANOVA.

## 2.6. Functional (flow tank particle capture) testing

For flow tests, mini-racks comprising six 20 × 7 cm sections of baleen (spaced 1 cm apart, as *in vivo* [11,17,22,23]) were clamped together on a metal rod suspended at the top of the 70 cm test chamber (900 mm$^2$ cross-sectional area) of a circulating flume [16]. This flume (i.e. flow tank) was filled with artificial seawater at 17°C flowing at 5–140 cm s$^{-1}$ ($n = 12$ trials for each flow speed). Buoyant (1 g cm$^{-3}$) latex beads with mean diameter 710 µm were added to the water (density approx. 15 000 particles m$^{-3}$) to determine particle capture rate per s. Trials were conducted with normal (unoiled) baleen versus baleen that had been submerged (minimum 24 h) in each of the eight kinds of oil (table 1). After the oil-treated baleen samples were removed from the oil, it took 10–15 min to clamp them into a rig to secure them within the flow tank. During this time, any surface oil may have dripped off the baleen, so additional oil was 'painted' on (brushed onto the baleen surface) just before the baleen samples were lowered into the water of the flow tank. Kinematic sequences were videotaped laterally and anteriorly underwater via illuminated digital endoscope (VideoFlex SD, Umarex-Laserliner, Arnsberg, Germany) recording JPEG images and AVI video (30 frames s$^{-1}$). Video was viewed frame-by-frame via GoPro Studio v. 2.5.7, with landmarks digitized via TRACKER v. 4.92. Kinematic analysis focused on the number of buoyant particles captured (=held to baleen fringes for greater than 3 s) relative to water flow speed and oil treatment.

# 3. Results

## 3.1. Oil adsorption results

Baleen gained 29–36+% mass ($\bar{x} = 32.3\%$ gain for all species, s.d. = 1.4, $n = 60$) when placed in water for one week (table 1), but gained relatively less (4–14%) mass ($\bar{x} = 11.2\%$ gain for all species, s.d. = 0.83, range 4–14%, $n = 340$) following a week's immersion in all oils. Air-dried baleen gained an average of 9.88% mass (for all species, s.d. = 0.91, range 7–13%, $n = 268$) following one week of immersion (submersion) in oil versus 17.1% for hydrated baleen (all species, s.d. = 1.14, range 13.0–20.4%, $n = 268$). Hydrated baleen gained more mass from oil exposure than air-dried baleen ($p = 0.04$). Baleen (air-dried or hydrated) gained less mass following submersion in any oil than baleen submerged in water alone for one week ($p = 0.02$; table 1).

Baleen samples from fin and humpback whales gained slightly more than bowhead and right whale baleen following 7 day exposure to oils (overall $\bar{x} = 10.3\%$ gain for dried and 17.6% gain for hydrated rorqual baleen, s.d. = 0.79 and 0.82, respectively, $n = 474$; overall $\bar{x} = 8.2\%$ gain for dried and 15.9% gain for hydrated balaenid baleen, s.d. = 0.44 and 0.57, respectively, $n = 480$), but there were no statistically significant differences ($p = 0.26$) among the four mysticete species tested here in terms of oil adsorption to baleen as measured by mass change.

Regarding the eight different oils that were tested (table 1), there were no statistically significant differences with regard to oil type ($p = 0.18$), although fish and possibly mineral oils were adsorbed slightly better. Results of mass increase from week-long exposure to (immersion in) each oil were as follows (all species combined): SAE-30 chain oil, $\bar{x} = 7.8\%$ gain in air-dried samples and $\bar{x} = 15.5\%$ gain in hydrated samples; GL-4 gear oil, $\bar{x} = 7.5\%$ gain in air-dried samples and $\bar{x} = 16.5\%$ gain in hydrated samples; SAE-40 engine motor oil, $\bar{x} = 7\%$ gain in air-dried samples and $\bar{x} = 16.8\%$ gain in hydrated samples; multipurpose lubricating oil, $\bar{x} = 10.3\%$ gain in air-dried samples and $\bar{x} = 17.3\%$ gain in hydrated samples; mineral oil, $\bar{x} = 10.5\%$ gain in air-dried samples and $\bar{x} = 20.1\%$ gain in hydrated samples, refined alkanes, $\bar{x} = 10.3\%$ gain in air-dried samples and $\bar{x} = 13\%$ gain in hydrated samples; cod liver oil $\bar{x} = 13\%$ gain in air-dried samples and $\bar{x} = 17.8\%$ gain in hydrated samples; tuna oil, $\bar{x} = 12.8\%$ gain in air-dried samples and $\bar{x} = 19.5\%$ gain in hydrated samples.

## 3.2. Histology results

Baleen samples that had been immersed in oil for 7 days (with surface oil allowed to run off but not wiped from the baleen surface) displayed limited tissue alteration and minimal accumulation of surface oil (figure 2). Macroscopic oil droplets (i.e. easily visible to the naked eye) measuring 2–8 mm in irregular splotches of varying size (overall $\bar{x} = 3.4$ mm, s.d. = 0.23, range 1.30–8.15 mm, $n = 140$) could be seen on 26.6% of baleen samples ($n = 420$). ImageJ analysis of photos revealed that these droplets covered 2–7% of the total photographed image (overall $\bar{x} = 4.8\%$, s.d. = 0.11, range 2–14%, $n = 140$). About half of the baleen samples (46.6%, $n = 420$) showed microscopic (less than 1 mm) droplets, typically in small assortments (dashed ovals in figure 2); some of these samples with microscopic oil also had large droplets (13.3% overall, $n = 420$). The microscopic droplets averaged 0.4 mm (400 µm) in size (s.d. = 0.08, range 0.1–1.0 mm, $n = 196$). Larger droplets were more frequently seen on balaenid (bowhead and right whale) baleen, whereas rorqual (fin and humpback whale) baleen specimens had slightly fewer large oil droplets yet more microscopic droplets (figure 2). An oily sheen was more frequently seen in rorqual baleen samples, although rorqual baleen is normally lighter in colour (typically grey or cream in colour) relative to the dark grey or black baleen of balaenids. Apart from this oily sheen there was no discoloration of the baleen. Oil of all types, both petroleum-based and biogenic, readily dripped from or was wiped or rinsed from baleen with little or no visible residue remaining.

In terms of potential tissue damage or other modification, multiple samples of all species (60%, $n = 420$) showed minor wrinkling, separation or displacement of outermost (cortical) keratin sheets (figure 2). These thin layers partially lifted and peeled away from underlying keratin sheets in some cases (9% overall, $n = 420$); this peeling was much more common in balaenid baleen, occurring in 34 of 210 samples (16%) but only 4 out of 210 rorqual samples (2%). In other cases (27%, $n = 420$), the uplifted keratin was not lifted or peeled off but merely wrinkled or bubbled up slightly. This was observed in only 5% (11 of 210) of balaenid samples but 49% (102 of 210) of rorqual samples, and was especially

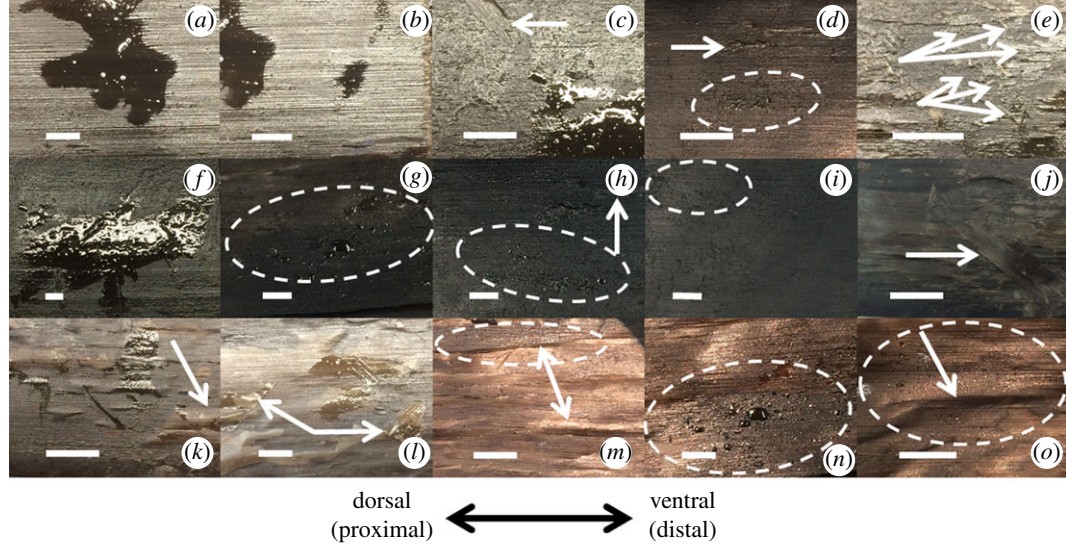

**Figure 2.** Micrographs of baleen samples immersed in oil for 7 days, all with the same orientation and scale bar = 1 mm (*a*−*e* North Atlantic right whale; *f*−*j* bowhead whale, *k,l* humpback whale, *m*−*o* fin whale), showing macro- and microscopic (within dashed circles) oil droplets and surface peeling effects explained in text. Arrows indicate where thin keratin sheets have loosened or separated from underlying matrix.

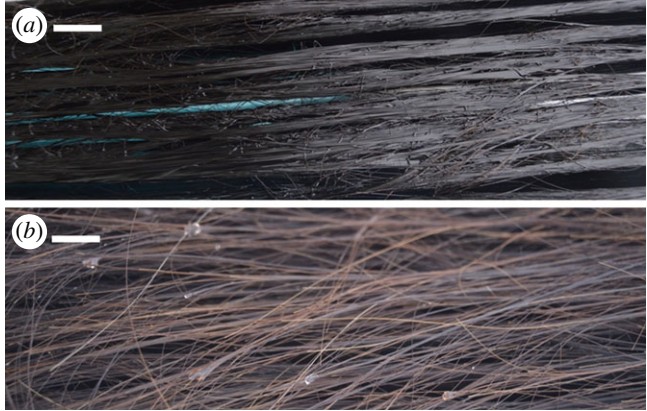

**Figure 3.** When submerged in motor oil (*a*), free fringes of baleen from a North Atlantic right whale cohere and appear darker, with an oily, glistening sheen. However, the same fringes look different (*b*) and display no evident oil accumulation after 5 min in flowing seawater (less than or equal to 0.6 m s$^{-1}$), indicating that oil rinses rapidly from baleen. Scale bar = 1 cm.

common (69%, $n = 105$) in fin whale baleen (figure 2). No other physical alteration of baleen was evident upon visual inspection.

Baleen fringes were wholly unaffected by oil immersion as judged by their appearance during microscopic examination. Oil almost wholly dissociated from fringes in less than 10 min in flowing seawater (figure 3) or fresh water. Most (greater than 75%) surface oil was removed within 3–5 min. This was observed in water flowing as slowly as 0.2 m s$^{-1}$.

## 3.3. Structural (mechanical) testing results

Results of strength testing (figure 4) reveal little difference between oiled versus unoiled treatments. Although 3-point bending tests revealed significantly different ($p = 0.01$, $n = 340$) flexural strength of dried versus hydrated baleen (prior to its immersion in oil; figure 4), the presence of the oil made no appreciable difference. Whether plain or oiled, unhydrated baleen fractured following the application of just over 20 N of force ($\bar{x} = 20.62$ N, s.d. $= 0.13$, range 20.35–20.88, $n = 180$) and after bending an average of 11.2 ($n = 168$) mm (figure 4). By contrast, water-soaked baleen, oiled and non-oiled, never

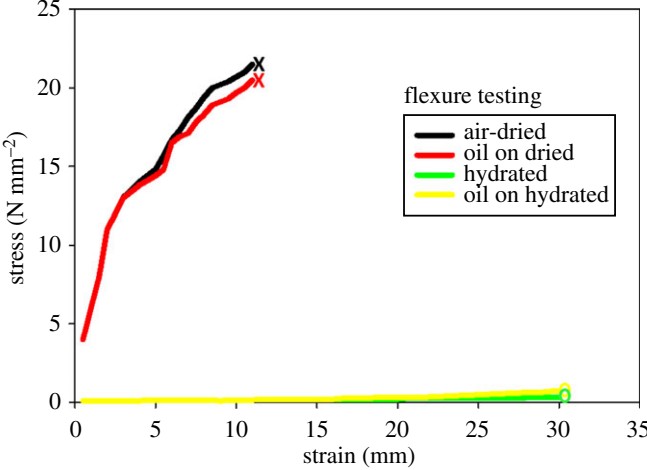

**Figure 4.** Maximal stress (in N mm$^{-2}$) recorded from 3-point flexural bending tests of oiled and unoiled baleen (previously dried in air or kept in flowing water for 7 days before 7 day oil exposure) shows that oil exposure makes no significant difference to baleen's material strength. Dry baleen is brittle and fractures with application of about 20 N mm$^{-2}$ whether or not it has been treated with oil; by contrast, wet baleen, with or without oil treatment, is highly flexible and reached the testing machine's displacement limit (30 mm deformation) with application of much lower forces (less than 1 N mm$^{-2}$).

failed (fractured) but reached the upper limit of the testing machine's range of measurement (30 mm displacement) after absorbing less than 2 N force ($\bar{x} = 1.37$ N, s.d. $= 0.18$, range 0.94–1.83 N, $n = 180$).

Oiled dry baleen was slightly but not significantly ($p = 0.47$, $n = 168$) more flexible than non-oiled dry baleen, with the oiled dry baleen bending and breaking at 19–20 N ($\bar{x} = 19.77$ N, s.d. $= 0.20$, $n = 84$) versus 20–22 N ($\bar{x} = 21.24$ N, s.d. $= 0.23$, $n = 84$) for plain dry baleen. The difference in maximum stress from strength testing results between oiled hydrated baleen ($\bar{x} = 0.62$ N, s.d. $= 0.09$, range 0.35–0.67, $n = 84$) versus unoiled hydrated baleen ($\bar{x} = 0.39$ N, s.d. $= 0.11$, range 0.31–0.72, $n = 84$) was also found to be not significant ($p = 0.26$, $n = 180$).

Despite these major differences in the flexibility of dried versus hydrated baleen samples (figure 4) terms of stress (applied force) and strain (resulting tissue displacement), there were no notable corresponding differences in flexibility when baleen was kept in oil compared to the control samples not kept in oil. Further, no differences in mechanical strength or flexibility were discerned between baleen tissues of the four tested species, nor of the eight types of oils tested.

## 3.4. Functional (particle capture) results

Particle capture from flow tank testing was not significantly ($p = 0.58$, $n = 180$) different with oiled or unoiled baleen (figure 5). Peak capture for oiled baleen ($\bar{x} = 6.35$ particles s$^{-1}$, s.d. $= 0.44$, $n = 84$) occurred at 1.2 m s$^{-1}$, whereas peak capture for unoiled baleen ($\bar{x} = 6.2$ particles s$^{-1}$, s.d. $= 0.41$, $n = 84$) was observed with water flow at 1.0 m s$^{-1}$. At lower flow speeds (less than 0.8 m s$^{-1}$) oiled baleen captured particles slightly but not significantly better than unoiled baleen ($\bar{x} = 3.47$ oiled versus 3.21 particles s$^{-1}$ unoiled, s.d. $= 0.12$ and 0.15, respectively, $p = 0.21$, $n = 84$); at higher speeds unoiled baleen captured slightly more particles than oiled baleen ($\bar{x} = 6.13$ unoiled versus 5.79 particles s$^{-1}$ oiled, s.d. $= 0.18$ and 0.16, respectively, $p = 0.39$, $n = 84$; figure 5). However, shortly (within 3–5 min) after baleen samples were exposed to flow nearly all oil had rinsed off the baleen fringes (figure 3), leading to no appreciable (=statistically significant) differences in baleen's filtering function. Baleen was found to capture small, buoyant plastic microspheres quite effectively whether or not oil remained present on the surface of baleen plates or fringes (figure 5).

Both oiled and unoiled baleen share similar trajectories of particle capture, with the most particles caught at flow speeds of 1.0–1.2 m s$^{-1}$. These flow speeds correspond to documented swim speeds of tagged whales during feeding, including balaenids [24,25] and rorquals [26–28]. Particle capture dropped slightly at higher flow speeds, most likely due to increased filter porosity [14] due to waving fringes and breakdown of the mat of entangled fringes [16,29].

There were no differences in particle capture experiment results from baleen of different species or by the types of oil applied to the baleen. The heavier gear and motor oil appeared slightly stickier

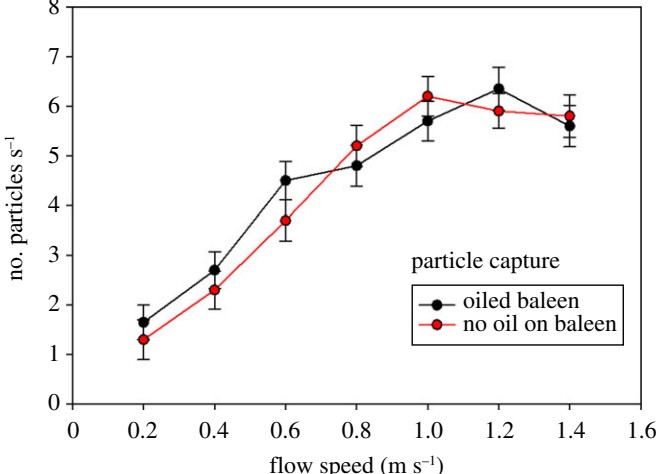

**Figure 5.** Particle capture (mean $\pm$ s.d.) during flow tank testing reveals no significant differences between oiled and unoiled baleen. For both treatments, peak particle capture (approx. 6 particles s$^{-1}$) occurred during flow speeds of 1.0–1.2 m s$^{-1}$.

(tackier to the touch), but this was not backed up by any statistical difference in particle capture by oil type ($p = 0.44$, $n = 84$). By contrast, it is possible that the lighter chain and multipurpose lubricating oils were more quickly rinsed from baleen than were other oils subjected to flowing water, but given that all oils were readily (i.e. within 3–5 min) rinsed from baleen plates and fringes in the flow tank study, no attempts to quantify the clearing time were made. Oil droplets were not found to be captured by the tangled mat of baleen fringes.

## 4. Discussion

Based on our results and initial power analysis (to determine minimum sample size with which to adequately evaluate our null hypothesis with 80% power and significance at the 0.05 level), our null hypothesis was supported: we found no appreciable statistical differences in the structural or functional properties of baleen following long-term (=one week) oil exposure. We did, however, detect slight surface changes in tissue appearance as described below.

Baleen is highly oleophobic and does not readily adsorb oils on its surface, even after complete immersion in oil for 7 days. Modest mass change indicates limited oil (relative to water) absorption by baleen (table 1). Wet baleen gains less mass than dried baleen, presumably because the water repels oil. This fits with results of a previous study [20] which showed that baleen is hydrophilic and readily interacts with water. Indeed, a decrease in hydrated baleen's mass following week-long oil immersion suggests oil forces out previously absorbed water (table 1), though this potentially causes loosening and separation of exterior-most cortical keratin layers (figure 2). Long-term oil exposure may push infiltrated water deeper into internal, intertubular baleen even as water simultaneously expels surface oil (which has not yet been fully adsorbed), yet this appears not to weaken baleen's material strength or flexibility (figure 4). Surface oil is poorly adsorbed and is easily and quickly rinsed by water flow, especially from waving fringes (figure 3). Oil does not appreciably change particle capture (figure 5). There were no significant differences between baleen of different species tested, nor of oils—from baleen's standpoint, all oils are the same: hydrophobic and therefore unlikely to interact with or alter baleen's characteristics. Work is underway to establish detailed adsorption isotherms; preliminary trials indicate oil is adsorbed better by baleen as temperature increases. Experiments with heavier fuel oils and chemical oil dispersants are also ongoing.

Our findings cast doubt on the likelihood of oil-contaminated baleen [3,5]. These results are somewhat unexpected given widespread fears of baleen fouling [1,3–5], although they fit earlier preliminary findings [3] that greater than 70% of surface oil is quickly washed from baleen and that less than 5% remains 24 h later. Keratin's oil-repelling ability [30] may even aid oil clean-up [31]. Our results are ultimately unsurprising given keratin's hydrophilic nature [17,20,32] and, upon reflection, mysticetes' diet of oily prey items [8,15]. Baleen's oleophobic nature makes both functional (ecological) and evolutionary sense in terms of whales collecting prey without their intra-oral filter clogging.

A crucial question concerns how whales and other marine organisms detect oils, either to avoid or seek them (if, in the latter case, oils provide a reliable sensory cue for the presence of plankton and fish). Mysticetes have a limited sense of smell [33], unlike toothed whales which apparently lack all olfactory ability. The extent to which mysticetes can detect and react to oils or oil-related compounds is unknown, but even a rudimentary olfactory capability may enable location of patchy prey [34]. Our results suggest this sensory ability is ultimately more important than the molecular-histological interaction between baleen and oil. Animal and petroleum oils have different aromatic compounds (hence fried chicken smells appealing, unlike gasoline) but share identical chemical properties of immiscibility with water or compounds with high water content, including baleen. Indeed, petroleum deposits originate from fossilized biogenic (planktonic) oil.

Even if fears of baleen fouling appear unfounded, the substantial risk of oil ingestion cannot be minimized [2]. Weak adsorption translates to oil being readily shed from baleen. This might initially seem advantageous; however, it might increase total risk of oil ingestion: rather than sticking to baleen, oil probably drips away from baleen. Shed oil potentially drips into the centre of the mouth where it could be more easily swallowed. Long-term bioaccumulation of petroleum oil within copepods [35,36] also increases ingestion risk in whale species where copepods or other oily zooplankton comprise much of the diet (especially bowhead and right whales but also the sei whale, *Balaenoptera borealis*). However, the potential risk of oil ingestion is mitigated by the fact that oil adheres poorly to baleen and does not have a chance to accumulate. By contrast, the efficiency with which baleen captured plastic microspheres in our experiments—and the way plastic debris, unlike oil, is readily trapped on and within baleen filter—suggests substantial risk to mysticetes from plastic pollution [37,38], which may thus ultimately pose a greater fouling risk (impeding the proper filtration necessary for feeding) and ingestion threat to whales than oil.

# 5. Conclusion

Our experiments reveal that baleen is hydrophilic and oleophobic, shedding rather than adsorbing oil. Second, we found that oil is easily rinsed by flowing seawater, minimizing the danger of baleen fouling. Third, our data show that oil does not significantly affect baleen's flexibility, strength or capture of particles. Fourth, our histological investigation revealed slight wrinkling and peeling of baleen's surface keratin layers caused by prolonged oil exposure. Fifth and finally, we conclude that ingestion of microplastics may pose a greater risk to whales than ingestion of oil. This conclusion is based on our findings of poor adsorption of oil to baleen, whereas we found (from our particle capture experiments) that the baleen filter is highly effective in trapping and accumulating small plastic particles, such that plastic is both more likely to clog the baleen filter and also perhaps more likely to be ingested by whales than oil.

Ethics. No animals were harmed for this research. All baleen tissue was obtained in accordance with applicable statutes under U.S. NOAA/NMFS Permits 17350 and 18786.
Data accessibility. All data are in a supplemental database file published online with this paper.
Authors' contributions. All authors collected/analysed data. S.M.B. and A.I.C. began the oil adsorption and mechanical testing experiments. A.J.W. expanded the adsorption and mechanical experiments and conducted histological and particle capture tests. A.J.W. wrote the draft and prepared figures. All authors edited the manuscript and gave final approval for publication.
Competing interests. The authors declare no competing interests.
Funding. This research did not receive any specific grant from funding agencies in the public, commercial, or not-for-profit sectors. All funding came from Hampden-Sydney College.
Acknowledgements. We thank I. Robertson and J. Jenkins for technical and laboratory assistance; T. Sformo, M. Moore, J.C. George, C. Mayo and G. Shigenaka for ideas; and T. Pitchford, W. McLellan, B. Adams, H. Brower and Q. Harcharek for access to baleen specimens. We thank two anonymous reviewers and Associate Editor Denise Greig for many helpful recommendations that greatly improved this manuscript.

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
