## [Reviewer comments · Royal Society Open Science]

Review History

RSOS-182194.R0 (Original submission)

Review form: Reviewer 1 (Nicholas Pyenson)

Is the manuscript scientifically sound in its present form?

Yes

Are the interpretations and conclusions justified by the results?

Yes

Is the language acceptable?

Yes

Is it clear how to access all supporting data?

Yes

Do you have any ethical concerns with this paper?

No

Have you any concerns about statistical analyses in this paper?

No

Recommendation?

Accept with minor revision (please list in comments)

Comments to the Author(s)

Werth et al. contribute an important, timely, and thorough study to address a basic question mostly unaddressed by anyone: does petroleum-based oils affect whale baleen? Surprisingly, no one has really tested this idea, and I applaud Werth et al. for a robust effort in mechanics, experimental design, and ecological context. I found a lot to like in this study, and see it as a fundamental finding that will garner many citations in the future -- after all, there will only be more whales mired in human-generated oil spills, not fewer (unfortunately).

I actually have no major issues with the manuscript in terms of structure or execution. The paper's clearly thought out, the materials are appropriately sourced and handled, and the authors provided a clear window into their thought process about mechanical tissue properties, and how to test for a signal. Their findings are interesting and noteworthy for a broad readership.

I don't need to see this manuscript again; just minor issues that should be corrected in their revisions:

Abstract:

There needs to be a sentence about what baleen is, what it's made out of, and why anyone (biologist or not) might care about. I realize that's a hard sentence to write, but it's important as the audience for RSOS is broad. There really needs to be more of a hook for the reader, in other words. I suggest "Baleen whales filter feed using..."

Scientific names should also be used instead of common names, especially in the abstract. Also, please use full words -- e.g., "and" and "or" -- instead of back slashes, which are brief and stylistically modern but not as informative as real words.

l. 17, Be specific -- species of whale. Potentially confused with prey species.

MAIN TEXT

l. 25, Again, provide more context about what a mysticete is. Realize that many readers of RSOS might not even know that we're talking about whales! I also think this is a key place to expand on that first sentence in the abstract describing the composition of baleen -- what is it, who has it, what does it do. It needs not be long, but it does need to be here, at this point in the manuscript.

l. 37, *Eubalaena* spp., period missing after "app"

l. 39, Family is not capitalized. And please use ">" instead of "+" for the words "in excess of"

l. 42, Please cite a general reference (e.g., the Kraus Urban Whale book) for why it is that right whales are endangered. A general reader might not know.

ll. 58-62, I don't think this formatting is necessary. It would be more seamless and artful, prose-wise, to have it inline with the text as full sentences. It wouldn't take much, and it would be worth the effort.

I. 70, Family is not capitalized.

I. 231, spell out 4 out of 120.

II. 260, 287. Again, backslashes are lazy and not communicating the nuance intended -- and or? Just and? Please be specific for mechanical strength and/or flexibility + cleansing and or clearing

I. 332. Or only for copepod-feeding mysticete? Isn't that just balaenids? Or do some mid-size rorquals feed on copepods? Specificity is warranted here.

L. 337, I am not a fan at all of checklist conclusions. The formatting is unappealing, it breaks from the rest of the manuscript, and it comes off as a bit lazy to me. I would appreciate the extra effort to write it out, in a few sentences. Doesn't take much, and it makes a big difference.

Review form: Reviewer 2

Is the manuscript scientifically sound in its present form?

Yes

Are the interpretations and conclusions justified by the results?

Yes

Is the language acceptable?

Yes

Is it clear how to access all supporting data?

Yes

Do you have any ethical concerns with this paper?

No

Have you any concerns about statistical analyses in this paper?

No

Recommendation?

Accept as is

Comments to the Author(s)

I found this to be a very interesting paper for a number of reasons, including the implications for the real-world environment the whales inhabit. The findings of this study are consistent with unpublished findings of a different research team that recently examined oil effects on baleen performance, and the implications discussed are consistent across both studies. I thought the use of multiple and qualitatively different endpoints was useful and appropriate, and strengthened the overall conclusions. The only quibble I have with this work is that the choice of petroleum test products was somewhat less than satisfying, in that all are refined/processed and not representative of the petroleum oils likely to be encountered in the environment (i.e., fuel or crude oils). I understand the issues and risks associated in working with, for example, crude oils, but an intermediate or heavy fuel oil would have provided an important reference that I'm not sure the other petroleum products provide. Similarly, given the ongoing controversies associated

with the use of chemical oil dispersants, inclusion of a representative product in the testing also would have been desirable. However: all other things being equal, this is a nice piece of work and well-thought out and implemented.

Decision letter (RSOS-182194.R0)

08-Apr-2019

Dear Dr Werth

On behalf of the Editors, I am pleased to inform you that your Manuscript RSOS-182194 entitled "Oil adsorption does not structurally or functionally alter whale baleen" has been accepted for publication in Royal Society Open Science subject to minor revision in accordance with the referee suggestions. Please find the referees' comments at the end of this email.

The reviewers and handling editors have recommended publication, but also suggest some minor revisions to your manuscript. Therefore, I invite you to respond to the comments and revise your manuscript.

- Ethics statement

- Data accessibility

If you wish to submit your supporting data or code to Dryad (<http://datadryad.org/>), or modify your current submission to dryad, please use the following link:
<http://datadryad.org/submit?journalID=RSOS&manu=RSOS-182194>

- Competing interests

- Authors' contributions

- Acknowledgements

- Funding statement

Because the schedule for publication is very tight, it is a condition of publication that you submit the revised version of your manuscript before 17-Apr-2019. Please note that the revision deadline will expire at 00.00am on this date. If you do not think you will be able to meet this date please let me know immediately.

- 1) A text file of the manuscript (tex, txt, rtf, docx or doc), references, tables (including captions) and figure captions. Do not upload a PDF as your "Main Document";
- 2) A separate electronic file of each figure (EPS or print-quality PDF preferred (either format should be produced directly from original creation package), or original software format);

- 3) Included a 100 word media summary of your paper when requested at submission. Please ensure you have entered correct contact details (email, institution and telephone) in your user account;
- 4) Included the raw data to support the claims made in your paper. You can either include your data as electronic supplementary material or upload to a repository and include the relevant doi within your manuscript. Make sure it is clear in your data accessibility statement how the data can be accessed;
- 5) All supplementary materials accompanying an accepted article will be treated as in their final form. Note that the Royal Society will neither edit nor typeset supplementary material and it will be hosted as provided. Please ensure that the supplementary material includes the paper details where possible (authors, article title, journal name).

on behalf of Dr Denise Greig (Associate Editor) and Professor Kevin Padian (Subject Editor)
openscience@royalsociety.org

Associate Editor Comments to Author (Dr Denise Greig):

This is a well constructed study that builds on the growing literature around balleen by considering how environmental pollution impacts balleen structure and function, and ultimately ingestion of pollutants. I would have loved to see some heavier oil and some oil dispersants tested as well, but maybe those will be part of future work. The manuscript is generally well written, but I got a bit lost in some of the different treatments and their terminology. I think this can be easily fixed relatively easily with a few definitions for a broader readership and consistent use of the words you use to describe your work. Below are some specific areas where I was confused.

Line 171. I was not sure what you meant when you said “submerged or painted”. Are these two separate oil treatments (1. submerged in oil for 7 days and 2. “painted” with oil) or just two ways to describe the same thing?

Line 186-189. This paragraph is confusing. I would prefer that you report the p values in the same sentences where you report the direction of the statistically significant effect. For example, hydrated baleen gained more mass than air-dried baleen ($p=...$) rather than comparing the significance of two separate results (i.e. to say that something is more significant than something else based on the p value of 0.02 versus 0.04 is probably not meaningful numerically or biologically). I’m also confused by the word adsorption. I thought these results were reporting on proportional mass change as a proxy for adherence. If adsorption and adherence are synonymous, I would pick one word and stick with it. It is also not clear whether relative and proportional mass gain are the same or two different measures. Is proportional when you compare the within baleen treatments and relative when you compare between treatments?

Line 226. Please add no “visible” residue....

Line 299-301. I think I am confused by Table 1 so I am not understanding how it relates to the text here. The text says the hydrated baleen decreased in mass after submersion in oil, but when I look at the table, the proportional changes are all above one so I thought that meant the mass increased. The title of the table is also confusing – it sounds like dried baleen was submerged in either water or oil, however the table appears to be baleen that was either dried or hydrated, and then all of the samples were submerged in oil. It is also unclear what the column labelled H2O means or why it is italicized.

Line 303. This seems to contradict the line before...does oil displace water or water displace oil?

Line 322. The “recognized olfactory ability” of whales is referenced in this sentence, but when I looked at the reference, it said that the mysticetes have a functional olfactory bulb (compared to odontocetes which have none), but are missing the genes used to actually “smell” and that they lack the ability to avoid “spoiled smells”. I wonder if they have another way to detect oils in their prey. Either way, please re-word this sentence to accurately reflect this earlier work.

Line 335. Your conclusion about microplastics seems contradictory to the sentence before where you suggest the ingestion risk from oil is enhanced because it does not adhere to the baleen and is more easily swallowed (that makes sense to me). I don’t disagree that microplastic ingestion is a global risk for these animals (especially the benthic feeding whales), I just am not sure how you are proposing that is more risky than oil ingestion. Or maybe ingestion risk is not exactly what you mean. Maybe exposure risk from ingestion? I am not sure what the risk is from having microplastics stuck in your baleen (exposure to chemicals on the plastics, baleen become less functional if full of microplastics)? I think you need to expand or clarify this assertion.

Reviewer comments to Author:

Reviewer: 1

Comments to the Author(s)

Werth et al. contribute an important, timely, and thorough study to address a basic question mostly unaddressed by anyone: does petroleum-based oils affect whale baleen? Surprisingly, no one has really tested this idea, and I applaud Werth et al. for a robust effort in mechanics, experimental design, and ecological context. I found a lot to like in this study, and see it as a

fundamental finding that will garner many citations in the future -- after all, there will only be more whales mired in human-generated oil spills, not fewer (unfortunately).

I actually have no major issues with the manuscript in terms of structure or execution. The paper's clearly thought out, the materials are appropriately sourced and handled, and the authors provided a clear window into their thought process about mechanical tissue properties, and how to test for a signal. Their findings are interesting and noteworthy for a broad readership.

I don't need to see this manuscript again; just minor issues that should be corrected in their revisions:

Abstract:

There needs to be a sentence about what baleen is, what it's made out of, and why anyone (biologist or not) might care about. I realize that's a hard sentence to write, but it's important as the audience for RSOS is broad. There really needs to be more of a hook for the reader, in other words. I suggest "Baleen whales filter feed using..."

Scientific names should also be used instead of common names, especially in the abstract. Also, please use full words -- e.g., "and" and "or" -- instead of back slashes, which are brief and stylistically modern but not as informative as real words.

l. 17, Be specific -- species of whale. Potentially confused with prey species.

MAIN TEXT

l. 25, Again, provide more context about what a mysticete is. Realize that many readers of RSOS might not even know that we're talking about whales! I also think this is a key place to expand on that first sentence in the abstract describing the composition of baleen -- what is it, who has it, what does it do. It needs not be long, but it does need to be here, at this point in the manuscript.

l. 37, *Eubalaena* spp., period missing after "app"

l. 39, Family is not capitalized. And please use ">" instead of "+" for the words "in excess of"

l. 42, Please cite a general reference (e.g., the Kraus Urban Whale book) for why it is that right whales are endangered. A general reader might not know.

ll. 58-62, I don't think this formatting is necessary. It would be more seamless and artful, prose-wise, to have it inline with the text as full sentences. It wouldn't take much, and it would be worth the effort.

l. 70, Family is not capitalized.

l. 231, spell out 4 out of 120.

ll. 260, 287. Again, backslashes are lazy and not communicating the nuance intended -- and or? Just and? Please be specific for mechanical strength and/or flexibility + cleansing and or clearing

l. 332. Or only for copepod-feeding mysticete? Isn't that just balaenids? Or do some mid-size rorquals feed on copepods? Specificity is warranted here.

L. 337, I am not a fan at all of checklist conclusions. The formatting is unappealing, it breaks from

the rest of the manuscript, and it comes off as a bit lazy to me. I would appreciate the extra effort to write it out, in a few sentences. Doesn't take much, and it makes a big difference.

Reviewer: 2

Comments to the Author(s)

I found this to be a very interesting paper for a number of reasons, including the implications for the real-world environment the whales inhabit. The findings of this study are consistent with unpublished findings of a different research team that recently examined oil effects on baleen performance, and the implications discussed are consistent across both studies. I thought the use of multiple and qualitatively different endpoints was useful and appropriate, and strengthened the overall conclusions. The only quibble I have with this work is that the choice of petroleum test products was somewhat less than satisfying, in that all are refined/processed and not representative of the petroleum oils likely to be encountered in the environment (i.e., fuel or crude oils). I understand the issues and risks associated in working with, for example, crude oils, but an intermediate or heavy fuel oil would have provided an important reference that I'm not sure the other petroleum products provide. Similarly, given the ongoing controversies associated with the use of chemical oil dispersants, inclusion of a representative product in the testing also would have been desirable. However: all other things being equal, this is a nice piece of work and well-thought out and implemented.

Author's Response to Decision Letter for (RSOS-182194.R0)

See Appendix A.

Decision letter (RSOS-182194.R1)

24-Apr-2019

Dear Dr Werth,

I am pleased to inform you that your manuscript entitled "Oil adsorption does not structurally or functionally alter whale baleen" is now accepted for publication in Royal Society Open Science.

on behalf of Dr Denise Greig (Associate Editor) and Kevin Padian (Subject Editor)
openscience@royalsociety.org

Associate Editor Comments to Author (Dr Denise Greig):
Associate Editor: 1
Comments to the Author:
(There are no comments.)

Reviewer comments to Author:

Appendix A

Response to reviewers

We thank the Associate Editor (Dr. Denise Greig) and two anonymous reviewers for their helpful comments in reviewing this manuscript. We have followed every one of their suggestions and made other changes based on their recommendations, which we believe has led to a much-improved text, table, and list of cited references.

We have added, at the end of the text, all of the end statements required by Royal Society Open Science, following the required headings and format.

As requested, we are submitting two versions of our revised manuscript: a clean copy and a copy **showing all changes highlighted in red.**

We will ensure that all image files are of proper quality and that a media summary is also submitted.

Our database file (including all raw data) will be published as an online supplement.

Below is our point-by-point response to each comment/recommendation.

Associate Editor Comments to Author (Dr. Denise Greig):

This is a well constructed study that builds on the growing literature around balleen by considering how environmental pollution impacts balleen structure and function, and ultimately

ingestion of pollutants. I would have loved to see some heavier oil and some oil dispersants tested as well, but maybe those will be part of future work. The manuscript is generally well written, but I got a bit lost in some of the different treatments and their terminology. I think this can be easily fixed relatively easily with a few definitions for a broader readership and consistent use of the words you use to describe your work. Below are some specific areas where I was confused.

Response: We have added mention that experimentation is ongoing with heavier fuel oils and chemical dispersants.

We have clarified the confusing sections about different treatments (both in the text and Table) and standardized definitions by referring only to adsorption and not adherence. Adsorption means adherence to a surface, but we agree that using both terms is potentially confusing to readers. We have also added some introductory information to make our paper more accessible to a wide audience.

Line 171. I was not sure what you meant when you said “submerged or painted”. Are these two separate oil treatments (1. submerged in oil for 7 days and 2. “painted” with oil) or just two ways to describe the same thing?

Response: We can see how this was confusing. We were trying to be concise but needed elaboration here, which we have added to explain that because oil often ran off baleen samples before flow tank trials could get underway, we “painted” oil on the samples to replicate what they were like when pulled from the oil in which they were submerged.

Line 186-189. This paragraph is confusing. I would prefer that you report the p values in the same sentences where you report the direction of the statistically significant effect. For example, hydrated baleen gained more mass than air-dried baleen ($p=...$) rather than comparing the significance of two separate results (i.e. to say that something is more significant than something else based on the p value of 0.02 versus 0.04 is probably not meaningful numerically or biologically).

Response: We agree this was confusing, and we have followed this advice in fixing it.

I'm also confused by the word adsorption. I thought these results were reporting on proportional mass change as a proxy for adherence. If adsorption and adherence are synonymous, I would pick one word and stick with it.

Response: Adsorption and adherence are basically synonymous, but we agree it is easier to use just one term. Thus we have eliminated mention of adherence. We hope this eliminates confusion.

It is also not clear whether relative and proportional mass gain are the same or two different measures. Is proportional when you compare the within baleen treatments and relative when you compare between treatments?

Response: Again, we can see how this was confusing. We have eliminated the term "proportional" which we were just using as a synonym for "relative."

Line 226. Please add no "visible" residue....

Response: We made this simple change.

Line 299-301. I think I am confused by Table 1 so I am not understanding how it relates to the text here. The text says the hydrated baleen decreased in mass after submersion in oil, but when I look at the table, the proportional changes are all above one so I thought that meant the mass increased. The title of the table is also confusing – it sounds like dried baleen was submerged in either water or oil, however the table appears to be baleen that was either dried or hydrated, and then all of the samples were submerged in oil. It is also unclear what the column labelled H₂O means or why it is italicized.

Response: We think the Table was the most confusing part of the paper and we spent much time revising it to add clarity and avoid confusion. First, we have moved the column labeled H₂O, which is (for reference) just weights of baleen submerged in water. The other columns show weights of baleen after submersion in a week in various oils following a different initial week in either air or water. This shows that wet baleen repels oil better than dried baleen. We have not only moved columns but changed the column headings and altered the heading/caption for this table for greater clarity. We think it makes much more sense now.

Line 303. This seems to contradict the line before...does oil displace water or water displace oil?

Response: We have expanded this sentence to clarify this. In short, we think both things are happening: wet baleen (with water that previously infiltrated the tissue) repels oil, and surface oil on baleen repels free water.

Line 322. The “recognized olfactory ability” of whales is referenced in this sentence, but when I looked at the reference, it said that the mysticetes have a functional olfactory bulb (compared to

odontocetes which have none), but are missing the genes used to actually "smell" and that they lack the ability to avoid "spoiled smells". I wonder if they have another way to detect oils in their prey. Either way, please re-word this sentence to accurately reflect this earlier work.

Response: There has been much study of the mysticete sense of smell, particularly in bowhead whales. We added another reference to this research. Many whale biologists believe whales can sense the dimethyl sulfide (DMS) that gives seafood its distinctive smell, and that this is how they detect krill or other zooplankton. Without going into detail here, we cite relevant research, but we also slightly expanded and clarified our points here in the Discussion.

Line 335. Your conclusion about microplastics seems contradictory to the sentence before where you suggest the ingestion risk from oil is enhanced because it does not adhere to the baleen and is more easily swallowed (that makes sense to me). I don't disagree that microplastic ingestion is a global risk for these animals (especially the benthic feeding whales), I just am not sure how you are proposing that is more risky than oil ingestion. Or maybe ingestion risk is not exactly what you mean. Maybe exposure risk from ingestion? I am not sure what the risk is from having microplastics stuck in your baleen (exposure to chemicals on the plastics, baleen become less functional if full of microplastics)? I think you need to expand or clarify this assertion.

Response: Yes, we did need to clarify this assertion, and to do so we expanded it as suggested. We do think that the risk of oil ingestion, while present, is minimal because oil does not seem to "stick" to baleen. In contrast, baleen readily traps microplastic, leading to serious potential for ingestion, along with greater potential for plastic fouling (clogging) of the filter relative to oil. We have expanded this with proper qualifying language.

Reviewer comments to Author:

Reviewer: 1

Comments to the Author(s)

Werth et al. contribute an important, timely, and thorough study to address a basic question mostly unaddressed by anyone: does petroleum-based oils affect whale baleen? Surprisingly, no one has really tested this idea, and I applaud Werth et al. for a robust effort in mechanics, experimental design, and ecological context. I found a lot to like in this study, and see it as a fundamental finding that will garner many citations in the future -- after all, there will only be more whales mired in human-generated oil spills, not fewer (unfortunately).

I actually have no major issues with the manuscript in terms of structure or execution. The paper's clearly thought out, the materials are appropriately sourced and handled, and the authors provided a clear window into their thought process about mechanical tissue properties, and how to test for a signal. Their findings are interesting and noteworthy for a broad readership.

I don't need to see this manuscript again; just minor issues that should be corrected in their revisions:

Abstract:

There needs to be a sentence about what baleen is, what it's made out of, and why anyone (biologist or not) might care about. I realize that's a hard sentence to write, but it's important as the audience for RSOS is broad. There really needs to be more of a hook for the reader, in other words. I suggest "Baleen whales filter feed using..."

Response: This is a good idea and we were happy to add an opening sentence just as suggested.

Scientific names should also be used instead of common names, especially in the abstract.

Response: This was easily remedied.

Also, please use full words -- e.g., "and" and "or" -- instead of back slashes, which are brief and stylistically modern but not as informative as real words.

Response: We are sorry that we had used so many slashes (largely to decrease the word count).

We have tried to eliminate nearly all slash use by adding proper conjunctions.

l. 17, Be specific -- species of whale. Potentially confused with prey species.

Response: We made this logical change.

MAIN TEXT

l. 25, Again, provide more context about what a mysticete is. Realize that many readers of RSOS might not even know that we're talking about whales! I also think this is a key place to expand on that first sentence in the abstract describing the composition of baleen -- what is it, who has it, what does it do. It needs not be long, but it does need to be here, at this point in the manuscript.

Response: As with the Abstract, we added a bit of necessary background here to aid all readers.

l. 37, *Eubalaena* spp., period missing after "app"

Response: We added the missing period.

l. 39, Family is not capitalized. And please use ">" instead of "+" for the words "in excess of"

Response: We changed the word family and we used a greater than symbol (>).

l. 42, Please cite a general reference (e.g., the Kraus Urban Whale book) for why it is that right whales are endangered. A general reader might not know.

Response: We added a new citation here: not the popular Kraus book but a similar article from Science.

ll. 58-62, I don't think this formatting is necessary. It would be more seamless and artful, prose-wise, to have it inline with the text as full sentences. It wouldn't take much, and it would be worth the effort.

Response: We thought the slimmer format was easier to scan, but we gladly made this change.

l. 70, Family is not capitalized.

Response: We omitted the word "family" here.

l. 231, spell out 4 out of 120.

Response: Done.

ll. 260, 287. Again, backslashes are lazy and not communicating the nuance intended -- and or? Just and? Please be specific for mechanical strength and/or flexibility + cleansing and or clearing

Response: We now spell all of these out, having eliminated slash marks throughout the revision.

l. 332. Or only for copepod-feeding mysticete? Isn't that just balaenids? Or do some mid-size rorquals feed on copepods? Specificity is warranted here.

Response: Good point—a rorqual (the sei whale) also feeds on copepods, as we now make clear.

L. 337, I am not a fan at all of checklist conclusions. The formatting is unappealing, it breaks from the rest of the manuscript, and it comes off as a bit lazy to me. I would appreciate the extra effort to write it out, in a few sentences. Doesn't take much, and it makes a big difference.

Response: We changed the numbered conclusions to a single paragraph as suggested.

Reviewer: 2

Comments to the Author(s)

I found this to be a very interesting paper for a number of reasons, including the implications for the real-world environment the whales inhabit. The findings of this study are consistent with unpublished findings of a different research team that recently examined oil effects on baleen performance, and the implications discussed are consistent across both studies. I thought the use of multiple and qualitatively different endpoints was useful and appropriate, and strengthened the overall conclusions. The only quibble I have with this work is that the choice of petroleum test products was somewhat less than satisfying, in that all are refined/processed and not representative of the petroleum oils likely to be encountered in the environment (i.e., fuel or crude oils). I understand the issues and risks associated in working with, for example, crude oils, but an intermediate or heavy fuel oil would have provided an important reference that I'm not

sure the other petroleum products provide. Similarly, given the ongoing controversies associated with the use of chemical oil dispersants, inclusion of a representative product in the testing also would have been desirable. However: all other things being equal, this is a nice piece of work and well-thought out and implemented.

Response: We are glad Reviewer 2 enjoyed the paper. Reviewer 2 is correct that working with crude or heavy oils is more difficult, but we are currently engaged in such experiments (and also with chemical oil dispersants), as we now make clear.